# Variational solution of the superconducting Anderson impurity model and the band-edge singularity phenomena

Teodor Iličin and Rok Žitko

*Jožef Stefan Institute, Jamova 39, SI-1000 Ljubljana, Slovenia and*
*Faculty of Mathematics and Physics, University of Ljubljana, Jadranska 19, SI-1000 Ljubljana, Slovenia*
(Dated: March 25, 2025)

We propose a set of variational wavefunctions for the sub-gap spin-doublet and spin-singlet eigenstates of the particle-hole symmetric superconducting Anderson impurity model. The wavefunctions include up to two Bogoliubov quasiparticles in the continuum which is necessary to correctly capture the weak-coupling asymptotics in all parameter regimes. The eigenvalue problems reduce to solving transcendental equations. We investigate how the lowest singlet state evolves with increasing charge repulsion $U$, transitioning from a proximitized state (a superposition of empty and doubly occupied impurity orbitals, corresponding to an Andreev bound state) to a local moment that is Kondo screened by Bogoliubov quasiparticles (Yu-Shiba-Rusinov state). This change occurs for $U = 2\Delta$, where $\Delta$ is the BCS gap. At this point, the band-edge effects make the eigenenergy scale in a singular way as $\Gamma^{2/3}$, where $\Gamma$ is the hybridization strength. Away from this special point, regular $\Gamma$-linear behavior is recovered, but only for $\Gamma \lesssim (U/2 - \Delta)^2/\Delta$. The singular behavior thus extends over a broad range of parameters, including those relevant for some quantum devices in current use. The singular state is an equal-superposition state with maximal fluctuations between the local impurity charge configurations. Accurately capturing the band-edge singularity requires a continuum model, and it cannot be correctly described by discrete (truncated) models such as the zero-bandwidth approximation or the superconducting atomic limit. We determine the region of parameter space where the second spin-singlet state exists: in addition to the whole $U < 2\Delta$ ABS region, it also includes a small part of the $U > 2\Delta$ YSR region for finite values of $\Gamma$, as long as some ABS component is admixed.

## I. INTRODUCTION

The Anderson impurity model with a superconducting bath is one of the paradigmatic models in the field of nanophysics, capturing the main properties of systems such as magnetic adsorbates on superconducting surfaces and quantum dots in contact with superconducting leads [1]. It is also relevant for hybrid semiconductor-superconductor quantum devices with potential applications in quantum information processing [2–4]. The Hamiltonian consists of a single impurity level $\epsilon_d$ with electron-electron repulsion $U$, hybridized with strength $\Gamma$ to a superconductor (SC) with a gap $\Delta$. Near half-filling, for $\epsilon_d \sim -U/2$, the system has up to three discrete low-lying many-particle eigenstates: one spin doublet as well as one or two spin singlets [5, 6]. In the doublet state, the dot is occupied by a single electron. How the lowest singlet state is obtained from this doublet by adding one particle to the system depends on the relative magnitudes of $U$ and $\Delta$. If $U$ is small, an electron or a hole can be added to the impurity level itself, incurring an energy cost of $\min(U + \epsilon_d, -\epsilon_d) \approx U/2$. If $U$ is large, it is energetically more favorable for an electron or a hole to be added to the superconducting bath in the form of a Bogoliubov quasiparticle, with an energy cost of $\Delta$. In the first case, the resulting singlet state is typically referred to as the proximitized level or the Andreev bound state (ABS), because the impurity wavefunction can be well approximated by a linear superposition of empty and doubly occupied states. In the second case, it is known as the Yu-Shiba-Rusinov (YSR) state, due to the similarity with the situation in the classical magnetic impurity Hamiltonian [7–9], but with quantum fluctuations generating a true singlet state between the impurity spin and the Bogoliubov quasiparticle from the static product state of anti-aligned spins. At the boundary between the two regimes, for $U \sim 2\Delta$, the wavefunction has a mixed character and no simple physical picture has been proposed to describe this situation yet.

The YSR singlet can be viewed as the superconducting analog of the Kondo singlet ground state in the Anderson impurity model with a normal-state bath. The Kondo problem is non-perturbative and characterized by a non-trivial low-energy scale, the Kondo temperature $T_K \sim \exp(-8\Gamma/\pi U)$. A detailed solution can be obtained numerically using impurity solvers such as Wilson's numerical renormalization group [10–13], or quantum Monte Carlo [14], but the essence of the problem can also be captured with remarkably simple variational wavefunctions that reproduce the exponential dependence of the characteristic energy scale on model parameters. The Yosida Ansatz focuses on the situation of half-filling in the Kondo Hamiltonian [15–17], the Appelbaum Ansatz is a generalization to the Anderson Hamiltonian [18], while the Varma-Yafet Ansatz also captures the physics of valence fluctuations [19]. Gunnarsson and Schönhammer further developed this approach for multi-orbital impurity problems, including with higher-order terms with multiple excitations in the bath, and remarking on the connection to the Brillouin-Wigner perturbation theory [20]. Variational methods have later been generalized to the superconducting case, first by Soda, Matsuura and Nagaoka for the Kondo Hamiltonian [21–

24]. While Kondo and Anderson impurity models with a normal-state bath are related in a simple way by the Schrieffer-Wolff transformation [25], the situation is more complex in a problem with a superconducting bath and depends on the ratio $U/\Delta$. For this reason, the variational wavefunctions for the Kondo Hamiltonian only describe the deep YSR limit and cannot capture the ABS physics. Later, a Varma-Yafet-type variational Ansatz was proposed for the $U = \infty$ limit of the Anderson model [26]. This approach cannot describe the ABS regime either.

In this work we propose variational wave-functions for the superconducting Anderson impurity model that includes terms of appropriate types for both ABS and YSR states. The Ansatz is hence applicable for all values of $U$. The resulting variational equations are integral equations that reduce to transcendental equations for the eigenvalues. We analyze the solutions with a particular focus on the non-trivial cross-over regime of $U \sim 2\Delta$, and discuss the singular behavior arising from the band-edge singularity physics, i.e., from the mixing between the discrete ABS-type states and the YSR-type states that involve the continuum of Bogoliubov quasiparticles. We determine the region in parameter space where the second singlet state exists inside the gap. We discuss the local properties of the states, their composition, as well as their spatial extent (specifically the localization length of the bound Bogoliubov quasiparticles).

## II. MODEL

We consider the Anderson impurity model with a superconducting bath, sketched in Fig. 1(a) and described by the following Hamiltonian:

$$H = H_{\mathrm{imp}} + H_{\mathrm{BCS}} + H_{\mathrm{c}} . \tag{1}$$

The first term describes the impurity orbital $d$:

$$H_{\mathrm{imp}} = \epsilon_d \sum_\sigma \left( n_\sigma - \frac{1}{2} \right) + U n_\uparrow n_\downarrow ,$$

where $\epsilon_d$ is the impurity level and $U$ the Coulomb repulsion, $n_\sigma = d_\sigma^\dagger d_\sigma$, and $d_\sigma$ is the annihilation operator for spin $\sigma$ on the impurity site. The following discussion assumes particle-hole symmetry, defined by the condition $\epsilon_d = -U/2$. The energy is shifted by $-\epsilon_d$ so that the energy of singly occupied isolated impurity is zero. The second term is the mean-field BCS Hamiltonian describing the superconducting bath:

$$H_{\mathrm{BCS}} = \sum_{k,\sigma} \xi_k c_{k,\sigma}^\dagger c_{k,\sigma} - \Delta \sum_k \left( c_{k,\uparrow}^\dagger c_{-k,\downarrow}^\dagger + \mathrm{h.c.} \right) .$$

Here $\xi_k = \epsilon_k - \mu$ is the single-particle dispersion (shifted by the Fermi level) of the conduction-band electrons and $\Delta$ is the superconducting gap. The third term couples

the dot with the lead:

$$H_{\mathrm{c}} = V \frac{1}{\sqrt{N}} \sum_{k,\sigma} \left( c_{k,\sigma}^\dagger d_\sigma + \mathrm{h.c.} \right) .$$

Here $N$ is the number of levels in the bath. The energy scale of hybridization is defined as $\Gamma = \pi \rho V^2$, where the density of states $\rho = 1/2D$ is chosen to be constant in the energy interval $[-D : D]$.

In the following, we will quantify the deviation from the special point $U = 2\Delta$ by the dimensionless parameter $\theta$ defined through

$$U = 2\Delta(1 + \theta) . \tag{2}$$

In the atomic limit ($\Gamma = 0$), the energy of the singlet, $E_S(\Gamma)$, is given by

$$E_S(0) = \min \left\{ \frac{U}{2}, \Delta \right\} \geq 0. \tag{3}$$

The two possibilities correspond to the two types of singlets states, ABS for $\theta < 0$ and YSR for $\theta > 0$. The nature of the states is depicted schematically in Fig. 1(b). We draw attention to the fact that for $\theta = 0$, the ABS-like state becomes degenerate with the bottom of the continuum of YSR-like spin singlets [6]. At finite hybridization, the energies will decrease due to the rearrangement of the quasiparticles in the system. We define the binding energy of the singlet state as

$$E_{\mathrm{bind}}(\Gamma) = E_S(\Gamma = 0) - E_S(\Gamma). \tag{4}$$

This is the quantity of main interest in this work.

## III. VARIATIONAL APPROACH

We now introduce variational wavefunctions for low-lying energy states in spin-doublet and spin-singlet subspaces. In order to validate the results, we make comparisons against high-precision reference results obtained using the numerical renormalization group (NRG) method [10, 13, 27–31].

### A. Doublet state

To describe the doublet state in the small hybridization limit ($\Gamma \to 0$), we use a variational Ansatz $|\phi\rangle = \sum_i |\phi_i\rangle$ with wavefunctions that contain up to one Bogoliubov quasi-particle:

$$
\begin{aligned}
|\phi_1\rangle &= a d_\uparrow^\dagger |\Phi_0\rangle , \\
|\phi_2\rangle &= \sum_k \mu_k \frac{1 + d_\uparrow^\dagger d_\downarrow^\dagger}{\sqrt{2}} \gamma_{k,\uparrow}^\dagger |\Phi_0\rangle , \\
|\phi_3\rangle &= \sum_k \nu_k \frac{1 - d_\uparrow^\dagger d_\downarrow^\dagger}{\sqrt{2}} \gamma_{k,\uparrow}^\dagger |\Phi_0\rangle ,
\end{aligned}
\tag{5}
$$

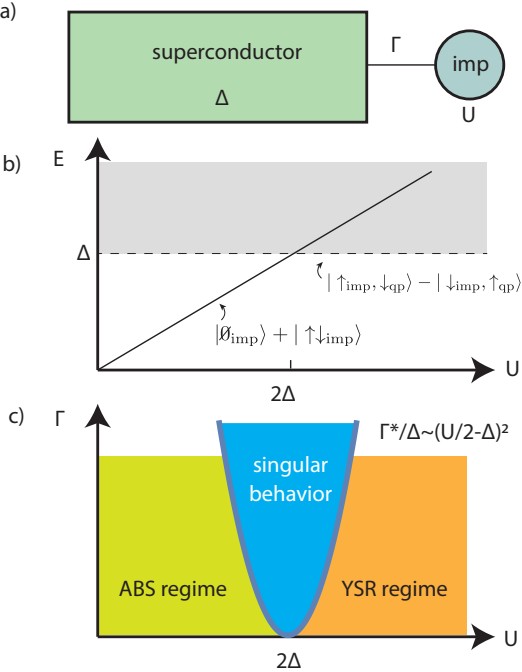

FIG. 1. a) Schematics of the model. b) Energy cost for creating low-energy spin-singlet excitations in the $\Gamma \to 0$ limit and the physical nature of the these states (proximitized states vs. exchange singlets). The shaded region corresponds to the continuum of singlets formed through the antiferromagnetic Kondo exchange coupling between the Bogoliubov quasiparticles and the impurity spin. c) "Phase" diagram of the problem. The singularity at the $U = 2\Delta, \Gamma = 0$ point influences a sizable region of the parameter space. This is the region of very strong mixing between the ABS and YSR components of the wavefunction, where the singlet state has properties different from either limiting type of behavior. It should be noted that at finite $\Gamma$ the variation as a function of $U$ is continuous: the region of singular behavior is a smooth cross-over between the ABS and YSR regimes. The maximal mixing occurs in the vicinity of $\theta = 0$, where the equal-superposition state with maximal fluctuations between impurity charge states is found (see Sec. VI). Along that vertical line the non-analytical behavior with fractional exponents is observed for all values of $\Gamma$.

where $|\Phi_0\rangle$ represents the state with the empty dot level and the SC leads in the BCS ground state:

$$|\Phi_0\rangle = |\hookleftarrow\rangle \otimes |\text{BCS}\rangle \,,$$

$$|\text{BCS}\rangle = \prod_k (u_k + v_k c^\dagger_{k,\uparrow} c^\dagger_{-k,\downarrow}) |0\rangle \,,$$

and $a, \mu_k, \nu_k$ are complex-valued variational parameters such that

$$\langle \phi | \phi \rangle = |a|^2 + \sum_k |\mu_k|^2 + \sum_k |\nu_k|^2 = 1. \qquad (6)$$

The quasiparticle operators $\gamma_{k,\sigma}$ are defined so that they diagonalize the BCS mean-field Hamiltonian:

$$\gamma_{k,\uparrow} \equiv u_k c_{k,\uparrow} - v_k c^\dagger_{-k,\downarrow} \,,$$

$$\gamma^\dagger_{-k,\downarrow} \equiv v_k c_{k,\uparrow} + u_k c^\dagger_{-k,\downarrow} \,.$$

Here $u_k = \sqrt{\frac{1}{2}\left(1 + \frac{\xi_k}{E_k}\right)}$, $v_k = \sqrt{\frac{1}{2}\left(1 - \frac{\xi_k}{E_k}\right)}$ and $E_k = \sqrt{\Delta^2 + \xi_k^2}$. The choice $u_k, v_k \in \mathbb{R}$ was made as only the case $\Delta \in \mathbb{R}$ is considered.

The energy of the doublet state ($E_D$) and the variational parameters $a, \mu_k, \nu_k$ are obtained by finding the extremal value of $f = \langle \phi | H | \phi \rangle - E_D(\langle \phi | \phi \rangle - 1)$, i.e., by taking derivatives with respect to $a^*$, $\nu_k^*$ and $\mu_k^*$:

$$E_D a = \frac{V}{\sqrt{2N}} \sum_k (u_k - v_k)\mu_k + \frac{V}{\sqrt{2N}} \sum_k (u_k + v_k)\nu_k \,, \qquad (7)$$

$$\left(E_D - E_k - \frac{U}{2}\right)\mu_k = a\frac{V}{\sqrt{2N}}(u_k - v_k) \,, \qquad (8)$$

$$\left(E_D - E_k - \frac{U}{2}\right)\nu_k = a\frac{V}{\sqrt{2N}}(u_k + v_k) \,. \qquad (9)$$

We switch to the continuum notation by relabeling $\xi_k \to x$ and $E_k, v_k, u_k, \mu_k, \nu_k \to E(x), v(x), u(x), \mu(x), \nu(x)$, where $x$ is now an energy variable. The normal-state DOS $\rho$ is constant, so that $(1/N)\sum_k \to \rho \int \mathrm{d}x$. After inserting Eqs. (8),(9) into Eq. (7) we obtain a transcendental equation for $E_D$:

$$E_D = -\frac{\Gamma}{\pi}\int_{-D}^{D} \frac{\mathrm{d}x}{E(x) + \frac{U}{2} - E_D} \,. \qquad (10)$$

This equation can be rewritten in closed form:

$$E_D = -\frac{\Gamma}{\pi}\left[ c_{\text{band}} + \frac{4(\Delta(1+\theta) - E_D)}{\sqrt{-(E_D - \Delta\theta)(E_D - \Delta(2+\theta))}} \times \right.$$
$$\left( \arctan\frac{\sqrt{-E_D + \Delta(2+\theta)}}{\sqrt{E_D - \Delta\theta}} - \right.$$
$$\left.\left. \arctan\frac{-D + \sqrt{D^2 + \Delta^2} - E_D + \Delta(1+\theta)}{\sqrt{(E_D - \Delta\theta)(-E_D + \Delta(2+\theta))}} \right) \right] \,,$$

where $c_{\text{band}} = 2\log\frac{\Delta}{D + \sqrt{D^2 + \Delta^2}}$. To investigate the $\Gamma \to 0$ asymptotics, we neglect $E_D$ in the right-hand side of the expression, as $\lim_{\Gamma \to 0} E_D = 0$. Since we work in the $\Delta \ll D$ limit, the expression can be further simplified by neglecting $\sqrt{D^2 + \Delta^2} - D$ in the second arctan term. After using the addition formula for the two arctan terms, we obtain the simplified expression for $E_D(\Gamma, \theta)$:

$$E_D \approx -\frac{\Gamma}{\pi}g_D(\theta) \,,$$

$$g_D(\theta) = \frac{4(1+\theta)}{\sqrt{-\theta(2+\theta)}}\arctan\sqrt{\frac{-\theta}{2+\theta}} \,. \qquad (11)$$

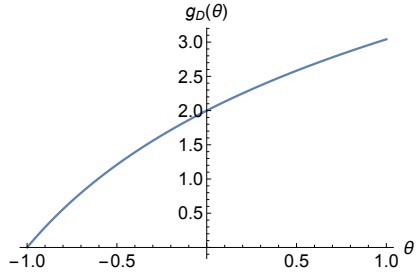

FIG. 2. Prefactor $g_D(\theta)$ in the $\Gamma$-linear dependence of the doublet eigenenergy, $E_D \approx -(\Gamma/\pi)g_D(\theta)$.

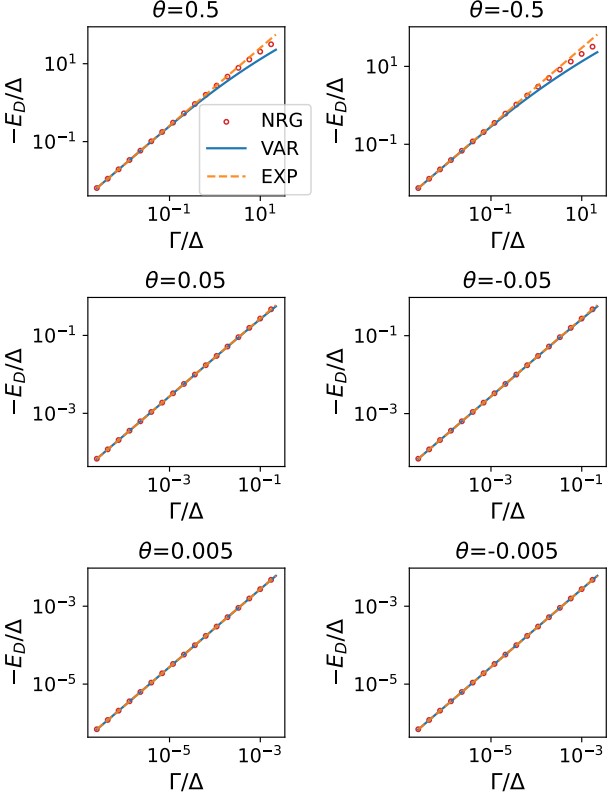

FIG. 3. Variational estimate of the doublet state energy $E_D$ vs. $\Gamma$ for different values of $\theta$. The solution of Eq. (10) is shown in blue, the expansion from Eq. (11) is shown in orange, while the NRG results are shown with red dots.

Note that this expression is real for both $\theta < 0$ and $\theta > 0$; in the latter case, the minus in front of $\theta$ is lost and arctan becomes arctanh. The function is continuous across $\theta = 0$ and goes to zero for $\theta = -1$ which corresponds to $U = 0$, see Fig. 2. In Fig. 3, we show that both the numerical solution of Eq. (10) and the low-$\Gamma$ expansion in Eq. (11) agree with the NRG data for small and intermediate values of $\Gamma$.

## B. Singlet state

We estimate the energy of the lowest-lying singlet in a similar way, by proposing a variational Ansatz $|\psi\rangle = \sum_i |\psi_i\rangle$ with the following components:

$$
\begin{aligned}
|\psi_1\rangle &= \frac{a_1}{\sqrt{2}}\left(1 + d_\uparrow^\dagger d_\downarrow^\dagger\right)|\Phi_0\rangle\,, \\
|\psi_2\rangle &= \frac{a_2}{\sqrt{2}}\left(1 - d_\uparrow^\dagger d_\downarrow^\dagger\right)|\Phi_0\rangle\,, \\
|\psi_3\rangle &= \sum_k \frac{\alpha_k}{\sqrt{2}}\left(d_\uparrow^\dagger \gamma_{k,\downarrow}^\dagger - d_\downarrow^\dagger \gamma_{k,\uparrow}^\dagger\right)|\Phi_0\rangle\,, \\
|\psi_4\rangle &= \sum_{k,k'} \frac{\beta_{k,k'}}{\sqrt{2}}\left(1 + d_\uparrow^\dagger d_\downarrow^\dagger\right)\gamma_{k,\uparrow}^\dagger \gamma_{k',\downarrow}^\dagger |\Phi_0\rangle\,, \\
|\psi_5\rangle &= \sum_{k,k'} \frac{\tau_{k,k'}}{\sqrt{2}}\left(1 - d_\uparrow^\dagger d_\downarrow^\dagger\right)\gamma_{k,\uparrow}^\dagger \gamma_{k',\downarrow}^\dagger |\Phi_0\rangle\,.
\end{aligned}
\tag{12}
$$

In the $\theta < 0$ regime, the ABS states $\psi_1$ and $\psi_2$ are the lowest-lying singlet eigenstates in the atomic limit. The same holds for the YSR state $\psi_3$ in the $\theta > 0$ regime. We find that in order to correctly capture the low-$\Gamma$ asymptotics all states that are directly coupled with the lowest-lying atomic-limit eigenstate need to be included in the Ansatz. Thus, the two-quasiparticle states $\psi_4$ and $\psi_5$ are required to correctly describe $E_S(\Gamma)$ in the YSR regime. This observation is key to obtaining the correct results [32]. We return to this point in Secs. V in VII.

We now note that due to the particle-hole (ph) symmetry of the problem, the variational equations in the even-parity and odd-parity sectors decouple. To see this, we define the ph transformation through

$$
d_\sigma \to -d_{\bar\sigma}^\dagger, \quad c_{k,\sigma} \to c_{-\bar k,\bar\sigma}^\dagger.
\tag{13}
$$

Here $\bar\sigma$ inverts spin from $\uparrow$ to $\downarrow$ and vice-versa. Furthermore, $\bar k$ corresponds to the momentum state that is conjugate with the momentum state $k$ with respect to the Fermi level, so that $\xi_k = -\xi_{\bar k}$. This furthermore implies $u_k = v_{\bar k}$. The Bogoliubov operators transform as

$$
\begin{aligned}
\gamma_{k,\uparrow} &\to u_k c_{-\bar k,\downarrow}^\dagger - v_k c_{\bar k,\uparrow} = v_{\bar k} c_{-\bar k,\downarrow}^\dagger - u_{\bar k} c_{\bar k,\uparrow} = -\gamma_{\bar k,\uparrow}, \\
\gamma_{-k,\downarrow}^\dagger &\to v_k c_{-\bar k,\downarrow}^\dagger + u_k c_{\bar k,\uparrow} = u_{\bar k} c_{-\bar k,\downarrow}^\dagger + v_{\bar k} c_{\bar k,\uparrow} = \gamma_{-\bar k,\downarrow}^\dagger.
\end{aligned}
\tag{14}
$$

The BCS wavefunction, $|\mathrm{BCS}\rangle$, is even. With these rules, we find that $\psi_1$ is a ph-even state, while $\psi_2$ is a ph-odd state. Furthermore, $\psi_3$ is ph-even if $\alpha_k = \alpha_{\bar k}$, or ph-odd if $\alpha_k = -\alpha_{\bar k}$. Finally, $\psi_4$ is ph-even if $\beta_{k,k'} = -\beta_{\bar k,\bar k'}$, odd if $\beta_{k,k'} = \beta_{\bar k,\bar k'}$, and $\psi_5$ is ph-even if $\tau_{k,k'} = \tau_{\bar k,\bar k'}$, odd if $\tau_{k,k'} = -\tau_{\bar k,\bar k'}$. Since for a ph-symmetric problem, all eigenstates have a definite parity, we can form spin-singlet states of two types: 1) ph-even states composed of $\psi_1$, $\psi_3$ with even $\alpha$, $\psi_4$ with $\beta_{k,k'} = -\beta_{\bar k,\bar k'}$ and $\psi_5$ with $\tau_{k,k'} = \tau_{\bar k,\bar k'}$ 2) ph-odd states composed of $\psi_2$, $\psi_3$ with odd $\alpha$, $\psi_4$ with $\beta_{k,k'} = \beta_{\bar k,\bar k'}$ and $\tau_{k,k'} = -\tau_{\bar k,\bar k'}$. In the absence of the particle-hole symmetry, $\langle H \rangle$ includes

terms of the type $a_1 a_2^*(\epsilon + U/2) + $ H.c., which describe mixing between the even and odd-parity sectors.

The problem of finding the energy of the singlet state is equivalent to the minimization of the expectation value of the Hamiltonian:

$$\langle\psi|H|\psi\rangle = \frac{U}{2}\left(|a_1|^2 + |a_2|^2\right) + \sum_k E_k|\alpha_k|^2 + \sum_{k,k'}\left(E_k + E_{k'} + \frac{U}{2}\right)\left(|\beta_{k,k'}|^2 + |\tau_{k,k'}|^2\right)$$
$$+ \frac{V}{\sqrt{N}}\left[a_1^*\sum_k \alpha_k(v_k + u_k) + a_2^*\sum_k \alpha_k(v_k - u_k) + \sum_{k,k'}\frac{1}{2}\alpha_k^*(u_{k'} - v_{k'})\beta_{k,k'} + \sum_{k,k'}\frac{1}{2}\alpha_{k'}^*(u_k - v_k)\beta_{k,k'} \right. \quad (15)$$
$$\left. + \sum_{k,k'}\frac{1}{2}\alpha_k^*(u_{k'} + v_{k'})\tau_{k,k'} + \sum_{k,k'}\frac{1}{2}\alpha_{k'}^*(u_k + v_k)\tau_{k,k'} + \text{c.c.}\right],$$

with a constraint originating from the normalization condition:

$$\langle\psi|\psi\rangle = \sum_i |a_i|^2 + \sum_k |\alpha_k|^2 + \sum_{k,k'}|\beta_{k,k'}|^2 + \sum_{k,k'}|\tau_{k,k'}|^2 = 1. \quad (16)$$

The energy of the singlet state $(E_S)$ and the variational parameters $a_i, \alpha_k, \beta_{k,k'}$ and $\tau_{k,k'}$ are obtained by finding the extremal value of $f = \langle\psi|H|\psi\rangle - E_S(\langle\psi|\psi\rangle - 1)$. Taking derivatives with respect to $a_1^*, a_2^*, \alpha_k^*, \beta_{k,k'}^*, \tau_{k,k'}^*$ results in the following equations:

$$\left(E_S - \frac{U}{2}\right)a_1 = \frac{V}{\sqrt{N}}\sum_k \alpha_k(v_k + u_k), \quad (17)$$

$$\left(E_S - \frac{U}{2}\right)a_2 = \frac{V}{\sqrt{N}}\sum_k \alpha_k(v_k - u_k), \quad (18)$$

$$(E_S - E_k)\alpha_k = a_1\frac{V}{\sqrt{N}}(v_k + u_k) + a_2\frac{V}{\sqrt{N}}(v_k - u_k)$$
$$+ \frac{V}{\sqrt{N}}\sum_{k'}(u_{k'} - v_{k'})\beta_{k,k'} + \frac{V}{\sqrt{N}}\sum_{k'}(u_{k'} + v_{k'})\tau_{k,k'}, \quad (19)$$

$$\left(E_S - \frac{U}{2} - (E_k + E_{k'})\right)\beta_{k,k'} =$$
$$\frac{V}{2\sqrt{N}}\alpha_k(u_{k'} - v_{k'}) + \frac{V}{2\sqrt{N}}(u_k - v_k)\alpha_{k'}, \quad (20)$$

$$\left(E_S - \frac{U}{2} - (E_k + E_{k'})\right)\tau_{k,k'} =$$
$$\frac{V}{2\sqrt{N}}\alpha_k(u_{k'} + v_{k'}) + \frac{V}{2\sqrt{N}}(u_k + v_k)\alpha_{k'}. \quad (21)$$

Note that the expression for $f$ needs to be properly symmetrized before the derivatives are taken. In the atomic limit $(V \to 0)$, the solution is given by $E_S = \min\{U/2, \min_k E_k\} = \min\{U/2, \Delta\}$, corresponding to $E_{\text{bind}} = 0$, as expected.

Again we switch to the continuum notation by relabeling $\xi_k \to x$ and $E_k, v_k, u_k, \alpha_k \to E(x), v(x), u(x), \alpha(x)$,.

The quasiparticle energy $E(x)$ is an even function, while the coherence factors are related by $u(-x) = v(x)$. The variational equations contain factors $s(x) = v(x) + u(x)$, which is an even function, and $a(x) = v(x) - u(x)$, which is odd. The equations decouple algebraically into two parity sectors, because $\int f(x)s(x)\mathrm{d}x$ filters out odd $f(x)$, while $\int f(x)a(x)\mathrm{d}x$ filters out even $f(x)$.

### 1. Even-parity sector

Here $a_1 \neq 0$ and $a_2 = 0$. We note that if $\alpha(x) = \alpha(-x)$, the even parity conditions are also automatically satisfied for $\psi_4$ and $\psi_5$ i.e. $\beta(x, x') = -\beta(-x, -x')$ and $\tau(x, x') = \tau(-x, -x')$. By expressing $a_1$, $\beta(x, x')$ and $\tau(x, x')$ from Eqs. (17), (20) and (21) and taking them into Eq. (19), we get:

$$(E_S - E(x))\alpha(x) = \frac{\Gamma}{\pi}\frac{s(x)}{E_S - \frac{U}{2}}\int_{-D}^{D}s(x')\alpha(x')\mathrm{d}x'$$
$$+ \alpha(x)\frac{\Gamma}{\pi}\int_{-D}^{D}\frac{\mathrm{d}x'}{N_2(x, x')} \quad (22)$$
$$+ \frac{\Gamma}{2\pi}s(x)\int_{-D}^{D}\frac{s(x')\alpha(x')}{N_2(x, x')}\mathrm{d}x',$$

where $N_2(x, x') = E_S - \frac{U}{2} - [E(x) + E(x')]$. Introducing auxiliary quantities $I_2(x) = \int_{-D}^{D}\frac{\mathrm{d}x'}{N_2(x, x')}$, $N_1(x) = E_S - E(x) - \frac{\Gamma}{\pi}I_2(x)$ and $\int_{-D}^{D}s(x)\alpha(x)\mathrm{d}x = C$, Eq. (22) can be rewritten in a more compact form:

$$\alpha(x) = \frac{\Gamma}{\pi}\frac{s(x)C}{\left(E_S - \frac{U}{2}\right)N_1(x)} + \frac{\Gamma}{2\pi}\frac{s(x)}{N_1(x)}\int_{-D}^{D}\frac{s(x')\alpha(x')\mathrm{d}x'}{N_2(x, x')}. \quad (23)$$

This is a Fredholm-like integral equation with a kernel that depends on the eigenvalue $E_S$. The kernel is not separable, which prevents one from simply integrating both sides and obtaining an equation for the eigenvalue $E_S$. We instead proceed by inserting the equation for $\alpha(x)$ into itself, more specifically into the second term on

the right-hand side:

$$\alpha(x) = \frac{\Gamma}{\pi} \frac{s(x)C}{\left(E_S - \frac{U}{2}\right) N_1(x)}$$
$$+ \frac{\Gamma^2}{2\pi^2} \frac{C}{E_S - \frac{U}{2}} \int_{-D}^{D} \frac{s(x)s^2(x')\mathrm{d}x'}{N_1(x)N_2(x,x')N_1(x')}$$
$$+ \frac{\Gamma^2}{(2\pi)^2} \int_{-D}^{D} \frac{s(x)s(x')^2 s(x'')\alpha(x'')\mathrm{d}x'\mathrm{d}x''}{N_1(x)N_2(x,x')N_1(x')N_2(x',x'')}.$$

This step is repeated iteratively, obtaining a series in $\Gamma$. We then multiply both sides with s(x) and integrate. $C$ cancels out and we obtain a transcendental equation for $E_S$:

$$E_S - \frac{U}{2} =$$
$$\frac{\Gamma}{\pi} \sum_{n=0}^{\infty} \left(\frac{\Gamma}{2\pi}\right)^n \int_{-D}^{D} \prod_{i \leq n} \frac{s^2(x_i)\mathrm{d}x_i}{N_1(x_{i-1})N_2(x_{i-1},x_i)N_1(x_i)},$$
$$(24)$$

where $N_1(x_{-1}) \equiv 1$ and $N_2(x_{-1},x_0) \equiv 1$.

Eq. (24) can be solved numerically by keeping some finite number of terms in the sum. In Fig. 4 the solutions with one and two terms are shown. The solutions of the equation with two terms give slightly better results for intermediate and large values of $\Gamma$. Going to higher orders in $\Gamma$ is computationally expensive, since high-dimensional quadratures need to be evaluated numerically, and is not expected to lead to much improvement. For large values of $\Gamma$ the variational Ansatz would instead need be extended with wavefunctions that include a larger number of Bogoliubov quasiparticles [33]. The approach described here works best for low and intermediate values of $\Gamma$, where it appears that even the lowest-order expansion in $\Gamma$ produces quite reliable results. The range of validity is further assessed in Sec. VIII. (See Ref. 34 for a strong-coupling approach to the same problem using a $1/\Gamma$ expansion.)

### 2. Asymptotic behavior of the lowest-lying singlet state

To lowest order (truncation to one term) Eq. (24) reads:

$$E_S = \frac{U}{2} + \frac{\Gamma}{\pi} \int_{-D}^{D} \frac{s^2(x)\mathrm{d}x}{E_S - E(x) - (\Gamma/\pi)I_2(x)}. \quad (25)$$

To obtain the asymptotic behavior of the binding energy in the $\Gamma \to 0$ limit, we rewrite this equation in terms of $E_{\mathrm{bind}}$. In the ABS regime, this means $E_{\mathrm{bind}} = \frac{U}{2} - E_S$, i.e.:

$$E_{\mathrm{bind}} = \frac{\Gamma}{\pi} \int_{-D}^{D} \frac{s^2(x)\mathrm{d}x}{E(x) - U/2 + E_{\mathrm{bind}} + (\Gamma/\pi)I_2(x)}. \quad (26)$$

Since $E(x) > \frac{U}{2}$ is always finite for $U < 2\Delta$, the part of the numerator that depends on $\Gamma$ can be neglected for

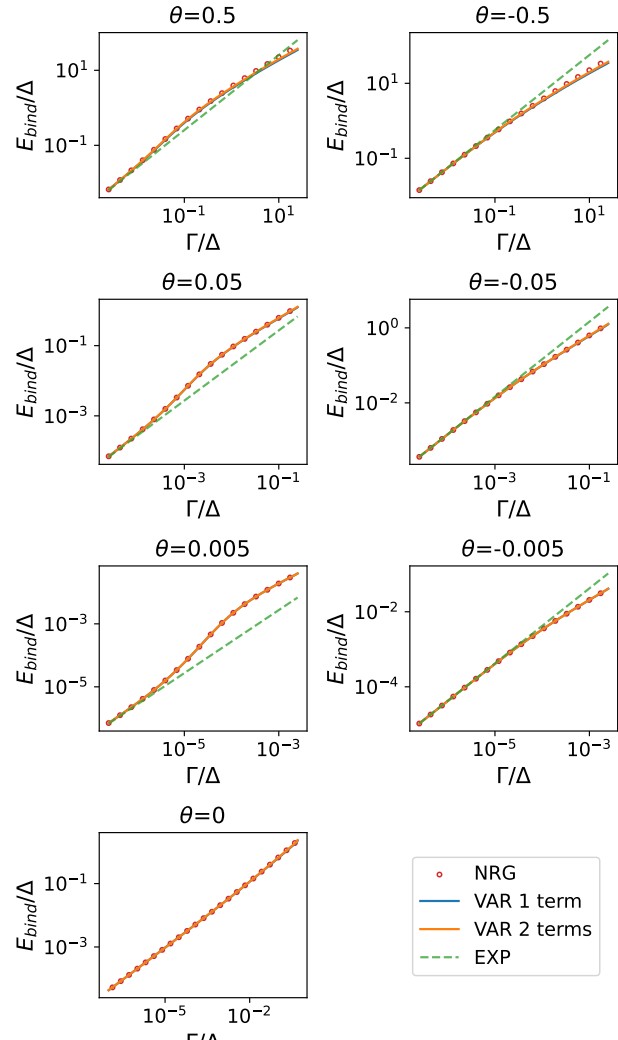

FIG. 4. Variational estimate of the binding energy $E_{\mathrm{bind}}$ vs. $\Gamma$ for different values of $\theta$ compared with the reference NRG results. Variational estimates are obtained by numerically solving Eq. (24), in which the high-order terms are neglected. The blue solid lines represent the solutions where only one term was kept, while the orange solid lines represent the solutions where the first two terms were included. Linear expansions from Eqs. (28), (30) are shown with dashed green lines. The NRG reference is shown with red dots.

$\Gamma \to 0$. This includes the binding energy on the right-hand side as we expect $E_{\mathrm{bind}} \sim \Gamma$ for small $\Gamma$. We thus get the expression for the binding energy in the $\theta < 0$ regime to the lowest order in $\Gamma$:

$$E_{\mathrm{bind}} \approx \frac{\Gamma}{\pi} \int_{-D}^{D} \frac{s^2(x)}{E(x) - \frac{U}{2}} \mathrm{d}x. \quad (27)$$

This can be evaluated to a closed-form expression:

$$E_{\text{bind}} \approx \frac{\Gamma}{\pi} 4(2+\theta)\left( \arctan\sqrt{-\frac{\theta}{2+\theta}} + \right.$$

$$\left. \arctan\frac{D - \sqrt{D^2+\Delta^2} + \Delta(1+\theta)}{\Delta\sqrt{-\theta(2+\theta)}} \right) + \frac{\Gamma}{\pi}c_{\text{Band}},$$

where $c_{\text{band}} = 2\log\frac{\Delta}{D+\sqrt{D^2+\Delta^2}}$. This can be simplified for $\Delta \ll D$ by neglecting $D - \sqrt{D^2+\Delta^2}$ in the second arctan term. We then use the arctan addition formula to obtain a compact expression:

$$E_{\text{bind}} \approx g_{\text{ABS}}(\theta)\frac{\Gamma}{\pi},$$

$$g_{\text{ABS}}(\theta) = 4\sqrt{-\frac{2+\theta}{\theta}}\arctan\sqrt{-\frac{2+\theta}{\theta}}. \quad (28)$$

The function $g_{\text{ABS}}$ is plotted in Fig. 5. It diverges for $\theta \to 0$, signaling the breakdown of the $\Gamma$-linear expansion in this limit.

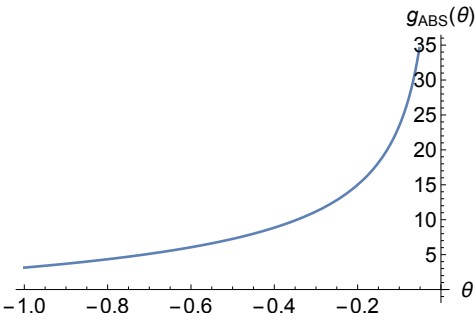

FIG. 5. Prefactor $g_{\text{ABS}}(\theta)$ in the $\Gamma$-linear dependence of the doublet eigenenergy.

In the YSR regime the binding energy is $E_{\text{bind}} = \Delta - E_S$, so Eq. (25) can be rewritten as:

$$E_{\text{bind}} = \frac{\Gamma}{\pi}\int_{-D}^{D}\frac{s^2(x)\mathrm{d}x}{E(x) - \Delta + E_{\text{bind}} + (\Gamma/\pi)I_2(x)}. \quad (29)$$

Here, the expansion cannot be done in such a straightforward manner because $\lim_{x\to 0}(E(x) - \Delta) = 0$ means that the denominator terms dependent on $\Gamma$ cannot be neglected. By surveying the NRG data and the numerical solutions of the variational equations, we notice that in the YSR regime, $-E_D$ and $E_{\text{bind}}$ match in the leading order of $\Gamma$, that is, $(E_{\text{bind}} - E_D) \sim \Gamma^2$. Therefore, up to first order in $\Gamma$, the binding energy in the YSR regime is:

$$E_{\text{bind}} \approx g_{\text{YSR}}(\theta)\frac{\Gamma}{\pi},$$

$$g_{\text{YSR}}(\theta) = g_D(\theta), \quad (30)$$

with $g_D(\theta)$ defined in Eq. (11). This observation about the relationship between the singlet and the doublet states in the weak coupling regime is particularly interesting in light of the recently explored relation between the singlet and the doublet in the strong-coupling regime [34].

### 3. Odd-parity sector

We now consider the odd-parity spin-singlet sector with $a_1 = 0$, $a_2 \neq 0$, and odd $\alpha(x)$. As before, the parity of $\alpha$ enforces the suitable parity of $\beta$ and $\tau$ via Eqs. (20), (21). After combining the minimization equations by analogy to the even-parity case, we get a similar equation for the energy of the odd-parity singlet state $E_2$:

$$E_2 - \frac{U}{2} =$$

$$\frac{\Gamma}{\pi}\sum_{n=0}^{\infty}\left(\frac{\Gamma}{2\pi}\right)^n\int_{-D}^{D}\prod_{i\leq n}\frac{a^2(x_i)dx_i}{N_1^{(2)}(x_{i-1})N_2^{(2)}(x_{i-1},x_i)N_1^{(2)}(x_i)}, \quad (31)$$

where the superscript (2) denotes that $E_2$ now takes the place of $E_S$ in the definitions of $N_1(x)$ and $N_2(x,x')$. The only essential difference between Eq. (31) and Eq. (24) is that $a(x)$ appears in place of $s(x)$.

Although it can be shown that the even-parity state always has lower energy, the odd-parity state can also be physically relevant, as it lies in the gap $(E_2 - E_D < \Delta)$ for certain values of $U$, $\Gamma$ [6]. More specifically, the second singlet state is always in the gap for $\theta < 0$ (i.e., in the ABS regime), whereas Eq. (31) dictates the existence of the second in-gap singlet state for $\theta > 0$ (in the YSR regime). The energy dependence of the odd-parity singlet state close to the $U = 2\Delta$ point is presented in Fig. 6(a) for a range of hybridization strengths.

The parameter space is thus partitioned into two subspaces depending on whether the odd-parity solution is in the gap, see Fig. 6(b). In fact, this line can serve as a possible definition for the boundary line separating the ABS and YSR regimes at finite $\Gamma$, since one of the characteristic features of the ABS regime is the possibility of holding zero, one or two Bogoliubov quasiparticles in the level (corresponding to the even-parity singlet, the spin-doublet, and the odd-parity singlet states, respectively). An equation for the boundary can be obtained by inserting the condition $E_2 = \Delta - E_D$ into Eq. (31), which, to the lowest order, reads:

$$\Delta + g_D(\theta)\frac{\Gamma}{\pi} = \frac{U}{2} +$$

$$\frac{\Gamma}{\pi}\int_{-D}^{D}\frac{a^2(x)\mathrm{d}x}{\Delta + g_D(\theta)\frac{\Gamma}{\pi} - E(x) - (\Gamma/\pi)I_2^{(2)}(x)}, \quad (32)$$

where $E_D$ is estimated using the approximate Eq. (11) and $E_2$ in $I_2^{(2)}(x)$ is replaced by $E_2 = \Delta + g_D(\theta)\frac{\Gamma}{\pi}$. The solution of this equation is shown as the blue line in Fig. 6(b). The equation of the boundary at large $\Gamma$ is significantly improved by also taking into account the subleading term in Eq. (31), see red line in Fig. 6(b).

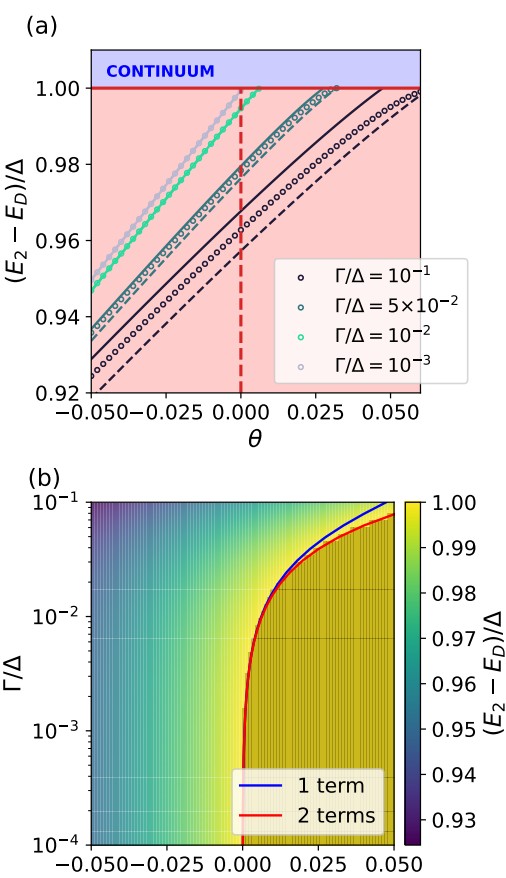

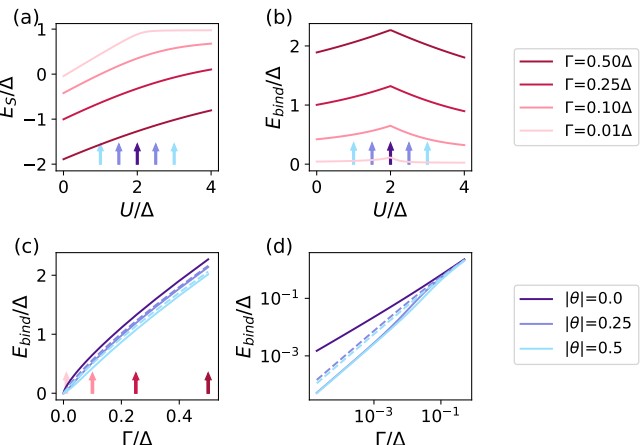

FIG. 7. Properties of the lowest singlet state as a function of $\Gamma$ and $U$ computed numerically using the NRG. (a) $U$-dependence of the singlet eigenenergy $E_S$ for a range of values of $\Gamma$. The values of $U$ ($\theta$) that correspond to curves in panel (c) are marked with blue arrows. (b) $U$-dependence of the singlet binding energy $E_{\text{bind}}$. (c) $\Gamma$-dependence of $E_{\text{bind}}$ for a range of values of the parameter $\theta$; full lines correspond to $\theta < 0$ (ABS regime), dashed lines to $\theta > 0$ (YSR regime). The values of $\Gamma$ that correspond to the curves in panels (a) and (b) are marked with red arrows. (d) Same data as in panel c, but on a log-log scale. Note the difference in the slope (i.e., different power-law exponents) between $\theta = 0$ and $\theta \neq 0$ for low $\Gamma$.

FIG. 6. Variational estimates and NRG reference results for the energy of the second in-gap singlet state shifted by the ground state energy, $E_2(\Gamma,\theta) - E_D(\Gamma,\theta)$, as a function of $\Gamma$ and $\theta$. (a) $E_2(\theta) - E_D(\theta)$ at different values of $\Gamma$. The variational estimates were obtained by taking into account the first or the first two terms in Eq. (31) and they are shifted by the variational estimate of the doublet-state energy from Eq. (11). The single-term solutions are shown with solid lines, while the two-term solutions are shown with dashed lines. The NRG data is shown with symbols (circles). (b) Energy of the second in-gap singlet state as a function of $\Gamma$ and $\theta$ as obtained by NRG (color coded). The shaded yellow region represents the part of the parameter space where the second singlet state is not in the gap, but rather dissolved in the Bogoliubov continuum. The analytical estimate of the boundary between the two regions, obtained by taking into account the first (two) term(s) in Eq. (31) is shown with the solid blue (red) line.

## IV. CROSS-OVER FROM ABS TO YSR REGIME

The variation of the eigenenergy of the lowest-lying singlet state with respect to $U$ and $\Gamma$, as obtained using the NRG, is presented in Fig. 7. Asymptotically, the binding energy $E_{\text{bind}}$ vs. $\Gamma$ is a power-law:

$$\frac{E_{\text{bind}}(\Gamma)}{\Delta} \sim \left(\frac{\Gamma}{\Delta}\right)^{\nu} . \tag{33}$$

For $\theta \neq 0$ we find that the asymptotic behavior is always linear, $\nu = 1$. The case $\theta = 0$ is exceptional: exactly at that point we find $\nu = 2/3$.

For finite $\Gamma$, we define an effective exponent through the logarithmic derivative

$$\nu_{\text{eff}} = \frac{\mathrm{d}\ln\frac{E_{\text{bind}}(\Gamma)}{\Delta}}{\mathrm{d}\ln\frac{\Gamma}{\Delta}} . \tag{34}$$

In Fig. 8 we plot the variation of $\nu_{\text{eff}}$ as a heat map in panel (a), and in the form of constant-$U$ 1D cross-sections in panel (b). The $\Gamma$-dependence of the binding energy crosses-over from the weak-coupling linear behavior to a different power-law regime at strong coupling (large $\Gamma$), with an intermediate region of $\nu_{\text{eff}} \approx 2/3$ for small values of $\theta$. The cross-over scales quadratically as

$$\Gamma_C/\Delta \sim (\theta/\Delta)^2 . \tag{35}$$

The change of behavior is much more pronounced for $\theta > 0$, where $\nu_{\text{eff}}$ shows a strongly non-monotonic variation.

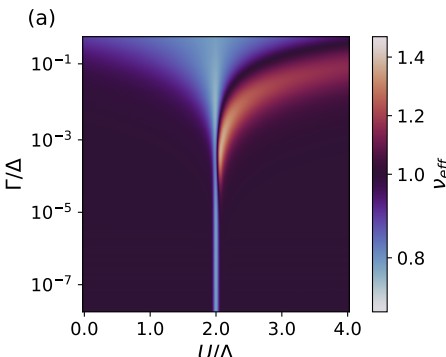

(a)

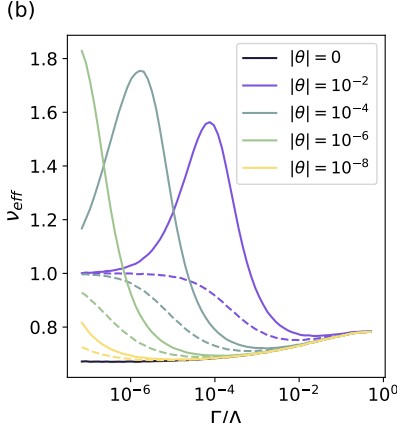

(b)

FIG. 8. Effective power-law exponent $\nu_{\text{eff}}$ describing the local $\Gamma$-dependence of the binding energy $E_{\text{bind}}$ as a function of $\Gamma$ and $U$. (a) Heat map in $(U, \Gamma)$ plane. Black corresponds to linear behavior. Blue color at $U/\Delta = 2$ corresponds to $\nu_{\text{eff}} \to 2/3$ asymptotic behavior at this singular point. (b) Constant-$U$ cross-sections. Full lines: $\theta > 0$, dashed lines: $\theta < 0$.

In Fig. 9 we plot the rescaled binding energy $E_{\text{bind}}(\Gamma)/E_{\text{bind}}(\Gamma_C)$ vs. $\Gamma/\Gamma_C$. For $\theta < 0$ there is some degree of universality, while for $\theta > 0$ the curves overlap less well, with a pronounced deviation as the $\theta = 0$ point is approached.

The exponent $2/3$ is due to the band-edge singularity generated by a band with an inverse square root divergence in the density of states at its edge, as is the case for the continuum of Bogoliubov excitations. To see this, we study the resonant level model for an impurity at energy $\epsilon_f$ hybridizing with a band with the density of states

$$\rho = \rho_0 (\epsilon/\epsilon_0)^\alpha \tag{36}$$

with $\alpha = -1/2$ in the energy interval $\epsilon \in [0 : D]$:

$$H_{\text{RLM}} = \epsilon_f f^\dagger f + \sum_k \epsilon_k g_k^\dagger g_k + \frac{V}{\sqrt{N}} \sum_k \left( f^\dagger g_k + \text{H.c.} \right). \tag{37}$$

We now set $\epsilon_0 = 1$ and $D = 1$ to simplify the expressions.

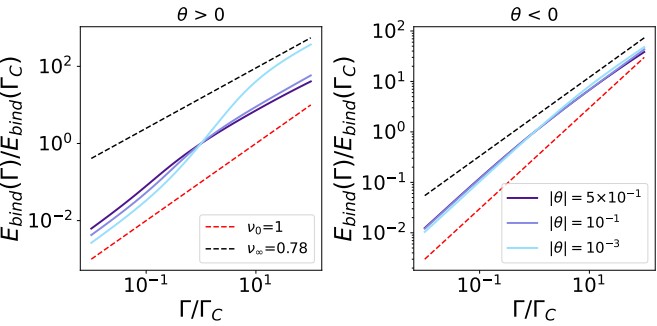

FIG. 9. Binding energy $E_{\text{bind}}(\Gamma)/E_{\text{bind}}(\Gamma_C)$ scaled as $\Gamma/\Gamma_c$. (a) YSR side for positive delocalization parameter $\theta$, (b) ABS side for $\theta < 0$.

The hybridization self-energy is

$$\Sigma(z) = V^2 \rho_0 \int_0^1 \frac{\epsilon^{-1/2}}{z - \epsilon} d\epsilon = \frac{2\text{arctanh}(1/\sqrt{z})}{\sqrt{z}}. \tag{38}$$

We write $z = \omega + i\delta$ and perform an analytical continuation to the real axis by taking the $\delta \to 0$ limit from above. For the real part of $\Sigma$ we find

$$\text{Re}\,\Sigma(\omega) = V^2 \rho_0 \frac{\sin(\arg \omega/2)}{\sqrt{|\omega|}} \times$$
$$\left[ \arg\left(1 - \frac{1}{\sqrt{\omega}}\right) - \arg\left(1 + \frac{1}{\sqrt{\omega}}\right) \right]. \tag{39}$$

For $\omega$ just below 0, this simplifies to

$$\text{Re}\,\Sigma(\omega) = -\frac{\Gamma}{\sqrt{-\omega}}, \tag{40}$$

where we defined $\Gamma = \pi \rho_0 V^2$. The bound state energy $E$ is given by

$$\epsilon_f + \text{Re}\,\Sigma(E) = E. \tag{41}$$

For $\epsilon_f = 0$, this gives $-\Gamma/\sqrt{-E} = E$ or

$$E = -\Gamma^{2/3}, \tag{42}$$

which concludes our derivation. [35]

We remark that an anomaly at $U = 2\Delta$ has been noted before in the context of the flat-band limit of Richardson model superconducting bath, but that calculation led to an exponent of $1/2$, see Eq. (68) in Ref. 36. The exponent $1/2$ actually corresponds to the behavior that is linear in $V$. In that case, the anomalous result at $U = 2\Delta$ results from a trivial degeneracy of discrete levels. Here, the exponent of $2/3$ results from the degeneracy of the discrete levels with the edge of the Bogoliubov continuum, and is a genuine band-edge singularity.

## V. VARIATIONAL WAVEFUNCTION PROPERTIES

The quality of the obtained variational wavefunction can be assessed through its ability to correctly describe

the nature of the eigenstate, in particular the electron distribution in the system. The reduced density matrix for the impurity level is calculated as the partial trace of the density matrix over all possible bath states:

$$\rho_{\text{imp}} = \langle \text{BCS} | \psi \rangle \langle \psi | \text{BCS} \rangle$$
$$+ \sum_{k,\sigma} \langle \text{BCS} | \gamma_{k,\sigma} | \psi \rangle \langle \psi | \gamma_{k,\sigma}^{\dagger} | \Phi_0 \rangle$$
$$+ \sum_{k,k',\sigma,\sigma'} \langle \text{BCS} | \gamma_{k',\sigma'}, \gamma_{k,\sigma} | \psi \rangle \langle \psi | \gamma_{k,\sigma}^{\dagger}, \gamma_{k',\sigma'}^{\dagger} | \text{BCS} \rangle \ ,$$

because up to two-quasiparticle excitations occur in the Ansatz. In the basis $\{|\hookleftarrow\rangle, |\uparrow\rangle, |\downarrow\rangle, |\uparrow\downarrow\rangle\}$, this can be written as

$$\rho_{\text{imp}} = \begin{bmatrix} \rho_0 & 0 & 0 & y \\ 0 & \rho_1 & 0 & 0 \\ 0 & 0 & \rho_1 & 0 \\ y & 0 & 0 & \rho_0 \end{bmatrix} ,$$

where

$$\rho_0 = \left( |a_1|^2 + |\beta|^2 + |\tau|^2 \right)/2,$$
$$\rho_1 = |\alpha|^2/2, \tag{43}$$
$$y = \left( |a_1|^2 + |\beta|^2 - |\tau|^2 \right)/2.$$

Here, we introduced the labels

$$|\alpha|^2 = \sum_k |\alpha_k|^2,$$
$$|\beta|^2 = \sum_{k,k'} |\beta_{k,k'}|^2, \tag{44}$$
$$|\tau|^2 = \sum_{k,k'} |\tau_{k,k'}|^2.$$

For p-h symmetric case, $\rho_0 \equiv \rho_2$.

To compute the matrix elements, we first express $\alpha(x)$ by neglecting the second, higher-order term in Eq. (23):

$$\alpha(x) = \frac{\Gamma}{\pi} \frac{s(x)}{\left( E_S - \frac{U}{2} \right) N_1(x)} C \ .$$

We then obtain equations for $a_1$, $\beta$ and $\tau$ by inserting this $\alpha(x)$ into Eqs. (17), (20) and (21). The constant $C = \int_{-D}^{D} \alpha(x') dx'$ can be determined from the normalization condition. The matrix elements are finally obtained by integrating the expressions for $|\alpha(x)|^2, |\beta(x,x')|^2$ and $|\tau(x,x')|^2$ in the continuum limit.

The excellent agreement of these expectation values with the reference NRG results, see Fig. 10, provides conclusive evidence confirming that the variational Ansatz accurately captures the main characteristics of the low-lying eigenstates.

An additional insight is offered by an analysis of the weights of the different terms in the variational wave-functions, see Fig. 11. The dominant part in the ABS regime is the local-pair state (high $|a_1|^2$). In the YSR regime, it is the exchange coupled state (high $|\alpha|^2$). This is fully expected and in line with the variation of $\rho_i$ presented in Fig. 10. We draw attention, however, to the two-quasiparticle terms ($|\beta|^2$ and $|\tau|^2$) that for $\theta > 0$ represent the subdominant terms in the $\Gamma \to 0$ limit.

The key role of the two-quasiparticle states in the YSR regime is further established by considering the contributions of the different terms in the expression for the total energy, Eq. (15). We plot those in Fig. 12 for the case of $\theta > 0$. More specifically, we show the contributions to the binding energy, $E_{\text{bind}}(\Gamma) = E_S(\Gamma = 0) - E_S(\Gamma)$, plotting their absolute values on the log-log scale in order to better reveal their magnitudes in the asymptotic regime. The dominant (linear in $\Gamma$) contribution to the binding energy is due to the hybridization with the two-quasiparticle states (dashed colored lines). This is facilitated by the lower energy of the two-quasiparticle states as opposed to the local singlet states ($a_1$-type states), due to $\Delta < U/2$ in the YSR regime. This explains why the two-quasiparticle terms must be included in the Ansatz, even if their total weight in the full wavefunction is very small.

## VI. EQUAL-SUPERPOSITION STATE GENERATING SINGULAR BEHAVIOR

We have anticipated that the singular behavior is due to the mixing between the discrete ABS-like states with the continuum of YSR-type singlet states. This can be seen quite directly by examining the variational wave-function for $\theta = 0$, see Figs. 10. There is an equal probability for the impurity level to be occupied by 0, 1 or 2 electrons ($\rho_0 = \rho_2 = \rho_{1\uparrow} + \rho_{1\downarrow} = 1/3$), thus this state corresponds to the case of maximal fluctuations between the different charge states. With increasing $\Gamma$, the two-quasiparticle components lead to a small deviation from a perfectly equal superposition.

## VII. LOCALIZATION OF QUASIPARTICLES IN THE YSR STATE

Since the solutions provides a momentum/energy re-solved wavefunctions, it is possible to study the localization of bound quasiparticles in the superconductor. Since in this work we aim for simplicity, we neglect the detailed lattice properties. We have already assumed a constant density of states and moved from the momentum space, $k$, to the energy space, $x$, tacitly assuming a linear dispersion. At this level of approximation, the real-space dependence (denoted by position $r$) can be simply obtained by a Fourier transformation from the $x$-space.

The behavior of the single-quasiparticle component of the wave function, parametrized by $\alpha(x)$ or $\alpha(r)$, is presented in Fig. 13. As expected, for weak hybridisation the quasiparticle is very weakly bound and thus delocalized over a large spatial region. As hybridization increases,

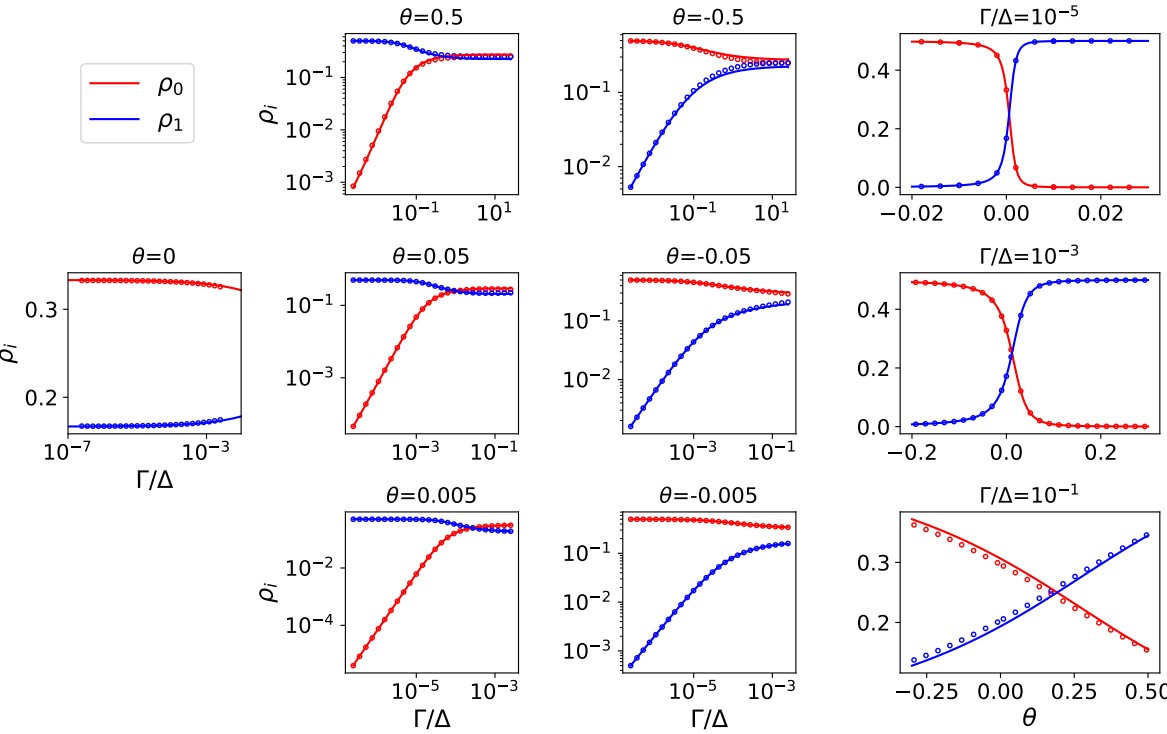

FIG. 10. Diagonal elements of the reduced density matrix vs. $\Gamma$ and $\theta$. Lines: variational estimates, symbols: NRG data.

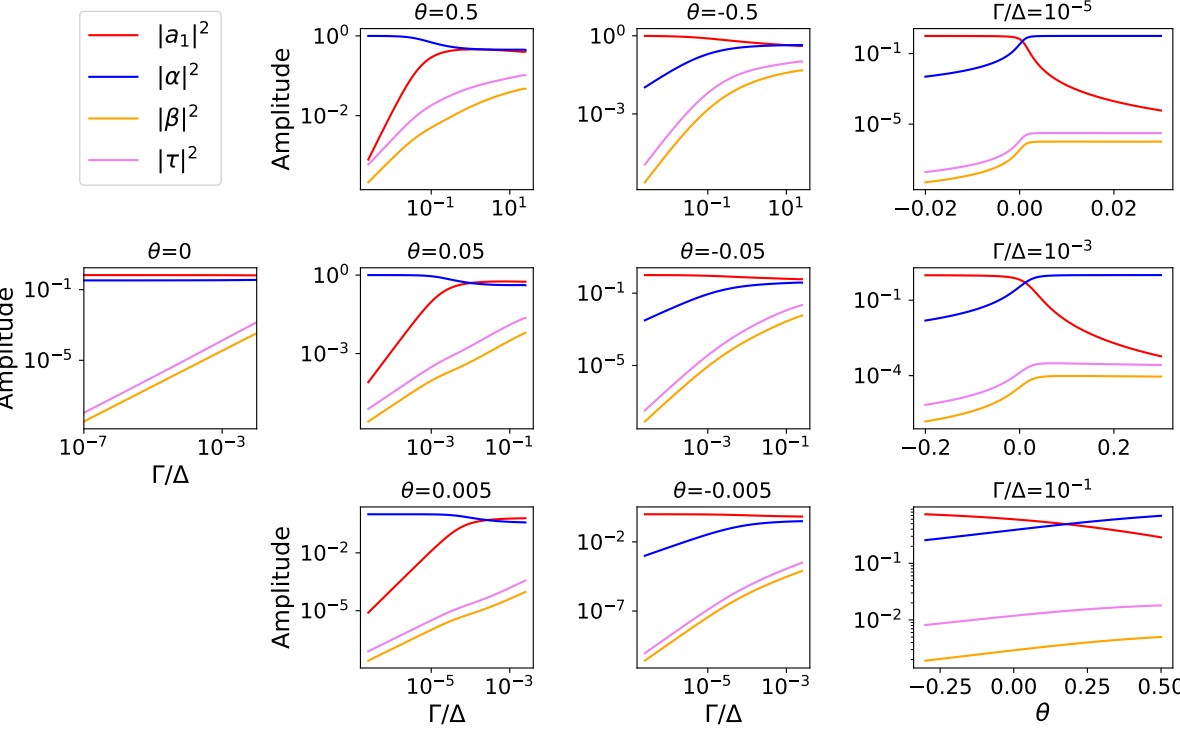

FIG. 11. Zero-, one- and two-quasi-particle state weights $|a_1|^2$, $\int |\alpha(x)|^2 dx = |\alpha(x)|^2$, $\int |\beta(x,x')|^2 dx dx' = |\beta|^2$, and $\int |\tau(x,x')|^2 dx dx' = |\tau|^2$ vs. $\Gamma$ and $\theta$.

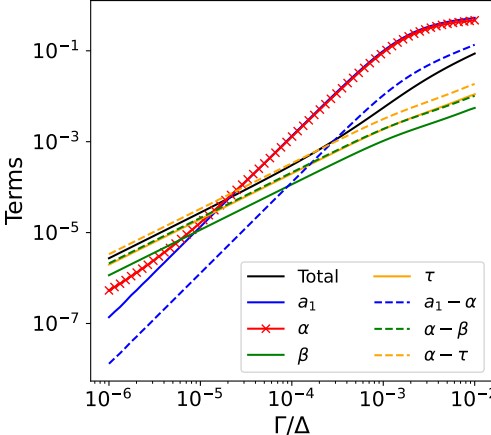

FIG. 12. Contributions to the singlet binding energy in the YSR regime, $\theta = 0.05$. We plot the absolute values of the different contributions to $E_{\text{bind}}(\Gamma) = \Delta - \langle\psi|H|\psi\rangle$. The line with markers corresponds to the dominant exchange singlet ($\alpha$ term), $\Delta - \langle\psi_3|H|\psi_3\rangle$. The thinner colored full lines correspond to other diagonal terms, $\langle\psi_i|H|\psi_i\rangle$, while the dashed lines correspond to the out-of-diagonal contributions, $\langle\psi_i|H|\psi_j\rangle$ with $i \neq j$. Finally, the black line is the total binding energy.

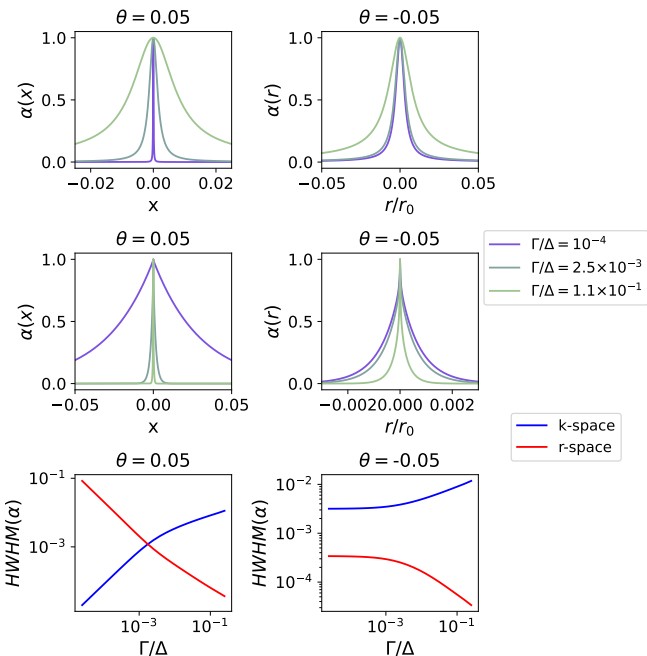

FIG. 13. Quasi-particle weight $\alpha$ in momentum and coordinate space. $r_0$ are arbitrary units. The bottom panels show the half-width at half-maximum of the peaks in the $\alpha$ distributions.

the quasiparticle localization length shrinks. The quasiparticle localization properties can be extracted from the width of the peaks in the $\alpha$ curves, see bottom panels in Fig. 13. On the YSR side ($\theta > 0$), the localization behaves as $1/\Gamma$, while on the ABS side ($\theta < 0$) the localization rate does not depend on $\Gamma$ asymptotically, which is in line with the character of the dominant parts of the wavefunctions. For larger $\Gamma$ we find a $1/\Gamma^\eta$ behavior with $\eta \approx 2/3$ as we enter the singular regime. This is yet another signature of the equal-superposition state.

Finally, we investigate the two-quasi-particle wavefunction. In the YSR regime, this component was found to explain the binding energy dependence on $\Gamma$. The weights $\beta$ and $\tau$ are large close to $x = 0$ and $x' = 0$ lines, or equivalently, close to the impurity position $r = 0$ in real space, see Fig. 14. One quasiparticle is thus strongly localized at the impurity site, while the other is delocalized over a large spatial region. This explains the singlet binding energy on the YSR side being nearly equal to the doublet binding energy, see Eq. (30): both receive dominant contributions from essentially the same processes, whereby the impurity electron hops to and from the bath, leading to order $\Gamma$ reduction in energy. In the singlet state the second quasiparticle is merely a spectator that ensures that the terms $|\psi_4\rangle$ and $|\psi_5\rangle$ are overall singlets. This is particularly striking in light of the recent observation that also in the strong-coupling limit, $\Gamma \to \infty$, the doublet and singlet states become essentially the same, up to the nearly decoupled quasiparticle [34].

## VIII. RELATION TO OTHER METHODS AND RANGE OF VALIDITY

Compared to the work of Rozhkov and Arovas, Ref. 26, our Ansatz includes the proper terms for the finite-$U$ situation at the particle-hole symmetric point. In particular, our singlet state Ansatz includes two local terms (even and odd) and two kinds of two-quasiparticle terms, while theirs has only one of each, omitting the double occupancy on the impurity site. Their doublet state is, however, more general and also includes two-quasiparticle terms, which we omit because we find that such terms do not bring about any new physical effects. The approach of Ref. 26 is limited to the $U = \infty$ limit and clearly does not apply to the $U = 2\Delta$ situation.

The variational approach in this work also shares some similarity with that of Meng et al., Ref. 6, who performed a perturbative expansion around the effective superconducting atomic-limit Hamiltonian. Their method starts from the action after integrating out lead electrons. The terms that correspond to the effective local Hamiltonian are identified, and the remaining terms are found to be the tunneling coupling between the dot and the leads beyond the lowest-order proximity effect. They perform the calculation to first order and resum the leading logarithmic divergencies, obtaining expressions for the energy shifts that resemble our expressions for the binding energies. However, the results differ. In particular, their equations for the energy corrections are only valid

$\theta = 0.05$

$\theta = -0.05$

FIG. 14. Two-quasi-particle weights $\beta$ and $\tau$ in momentum and coordinate space at $\Gamma/\Delta = 2.5 \times 10^{-3}$ and $|\theta| = 0.05$. $r_0$ are arbitrary units. The lines of constant $x_2$ and $r_2$, shown in the second and fourth columns, correspond to the indications in the first and third columns.

as long as the renormalized excitation energies are not too close to the gap edge, thus the low-$\Gamma$ asymptotics in the YSR regime cannot be described due to logarithmic divergences, in contrast to the variational method that becomes exact for small $\Gamma$.

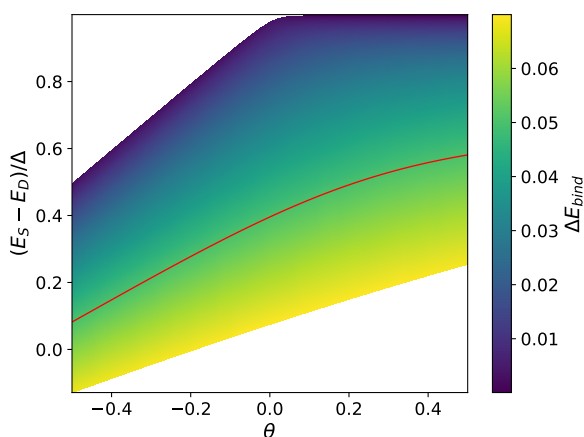

FIG. 15. Relative error in the singlet state energy, $\Delta E_{\text{bind}}$, vs. $\theta$ and $\Gamma$. On the vertical axis, $\Gamma$ is rescaled in terms of the excitation energy $E_S - E_D$ in order to indicate the position of the subgap excitation peak within the gap. The 5% error is shown with the red contour.

On the other hand, the method of Meng et al. has been found to match rather well the NRG results for the experimentally relevant intermediate parameter range ($\Gamma$, $U$ and $\Delta$ of comparable magnitude). It is thus of some interest to determine the range of validity of the variational method. In Fig. 15 we plot the relative error of the singlet energy as a function of $\theta$ and $\Gamma$, taking the NRG results as the reference. The $\Gamma$-dependence is converted to the excitation energy $(E_S - E_D)/\Delta$. We observe that for low $\Gamma$ (corresponding to $E_S - E_D = \min\{U/2, \Delta\}$) the error is indeed vanishing. In the range $E_S \sim E_D$, in vicinity of the singlet-doublet quantum phase transition, the errors are typically in the 10% range and they grow somewhat larger deeper in the YSR regime. We can therefore state that the methods are complementary: they both provide reliable results in the intermediate-couple regime, but in addition the variational method also describes well the weak-coupling regime, while the method of Ref. 6 also captures the strong-coupling regime beyond the singlet-doublet transition line.

In recent years, there have also been several developments that aim to provide an accurate description of the full impurity problem in terms of a small number of orbitals in a controlled way. One approach is to parametrize the bath hybridisation function in terms of multiple representative proximitized orbitals [37]. Increasing the number of orbitals leads to an improved level of approximation. Another approach (that remains to be generalized to the superconducting case) is that of natural orbitals [33], where the optimal ("active") set of orbitals is constructed based on the eigenstates of the one-particle density matrix. Such approaches are expected to approximately reproduce the singular behavior at $U = 2\Delta$ if a sufficient number of orbitals is used, although perhaps at large computational cost.

## IX. CONCLUSION

We have proposed variational wavefunctions for the lowest-lying eigenstates of the single-impurity Anderson model coupled to a superconducting bath. The key advantage of the variational solution is its ability to correctly include the continuum effects and the possibility of obtaining closed-form asymptotic expressions for the eigenvalues. This capability has allowed us to study the cross-over between the ABS and the YSR regimes of the excited singlet state, the non-analytical point at $(U = 2\Delta, \Gamma \to 0)$ and the resulting singular behavior in an extended region of the parameter space. We discovered that in this region the wavefunction is an equal-superposition state where the local charge state on the impurity state is maximally fluctuating. This is the result of band-edge singularity physics, leading to strong mixing between ABS-type and YSR-type terms. Given that the parameter range $U/2 \sim \Delta$ can be accessed in hybrid semi-super quantum devices, these platforms could serve as a testing ground to probe the predictions in full detail, e.g. using high-precision microwave spectroscopy [38].

This approach could be fairly easily extended in different ways, for example to the general case without the particle-hole symmetric, to a two-lead situation with a finite BCS phase bias (i.e., a quantum-dot Josephson junction problem), and to the presence of magnetic field (spin-splitting in the doublet state). It should also be possible to extend the method to multi-impurity and multi-orbital problems.

## ACKNOWLEDGMENTS

We acknowledge the support of the Slovenian Research and Innovation Agency (ARIS) under P1-0416 and J1-3008. We thank Luka Pavešič, Volker Meden, Jens Paaske, and Serge Florens for their comments.

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
