# Peer review of "Variational solution of the superconducting Anderson impurity model and the band-edge singularity phenomena"

_SciPost Physics, doi:SciPost Phys. 19, 006 (2025)_

## Round 1 · Referee Report · Anonymous (Referee 2) · 2025-6-6

Report
I think that this paper is well motivated and well placed in the context of previous variational approaches, as is detailed convincingly in the introduction and also in Sec. VIII. The ansatz is made plausible, and the variational optimization and the results derived are described very precisely. Results from the analytical and numerical evaluation are favorably compared with NRG data. The discussion of the band-edge singularity, i.e., the NRG result, Fig. 8, and the analytical result Eq. 42 is particularly interesting.
Concluding, this is a strong paper that should be published.
There are some optional and minor points the authors might take into account:
One could try to improve the very technical presentation, in particular in Sec. III B, by shifting some details to the appendix.
The dark green color in Fig. 6a is hard to distinguish from black.
In the sentence above Eq. 33, one may add "asymptotically, for Gamma/Delta -> 0, ..." for clarity.
It should be explicitly indicated (in the text and in the figure captions), which method is used (variational wave function or NRG) to get the results shown in Figs. 7 and 8, and how the heat map is constructed in practice.
Recommendation
Publish (easily meets expectations and criteria for this Journal; among top 50%)

---

## Round 1 · Referee Report · Anonymous (Referee 1) · 2025-6-6

Strengths
Weaknesses
Report
quasiparticles appearing in the correlated (Anderson-type) quantum impurity
coupled to the conventional (BCS-type) superconductor. They propose variational
wave-function approach to describe both, the Andreev or Yu-Shiba-Rusinov bound
states, which is valid for the arbitrary set of model parameters: Coulomb
repulsion (U), pairing gap ($\Delta$), hybridization strength ($\Gamma$). Specifically,
they focus on the half-filled impurity in the regime, U $\sim 2 \Delta$, where
changeover of the bound states occurs.
Details concerning construction of the doublet and singlet configurations are
presented in Sec. II - technicalities are convincingly discussed and numerical
results satisfactorily agree with the unbiased NRG calculations (Sec. IV).
Next, the authors analyze properties of their variational wave-functions,
inspecting matrix elements of the low-lying eigenstates (Sec. V). In particular,
they emphasize the maximal fluctuations between different charge-number states
in the case U = 2 $\Delta$. Finally the spatial profiles are investigated through
the Fourier-transformed momentum/energy resolved wavefunctions. Numerical
results (Fig. 13) indicate that upon increasing the hybridization strength
(Gamma) the quasiparticles tend to be more and more localized. In Sec. VIII
the authors confront their procedure with other methods, Refs. [26, 6, 37]
and clarify their limitations and/or complementary character.
In my opinion, the method proposed by T. Ilicin and R. Zitko could be of
potential use for studying various superconducting nanohybrid structures.
The manuscript is clearly written, analytical expressions seem to be
correct, validity of the computational data is carefully checked by
comparing them to NRG and other available methods. I thus recommend
the paper for publication in its present form.
Requested changes
None
Recommendation
Publish (surpasses expectations and criteria for this Journal; among top 10%)

---

## Round 2 · Author Response

The revised manuscript has improved presentation of result and the reference list has been updated with DOIs. The SciPost template is now being used.

---

## Round 2 · List of Changes

• Figures improved for readability (color schemes).
  • Sec IIB made more compact by shifting the discussion of symmetries to the introduction and moving the lengthy expression for the energy expectation value to an Appendix.
  • Provenience of data (NRG or variational method) indicated in all captions, details about postprocessing to obtain the heat map explained.
  • Bibliography improvements (DOIs added, missing volume and article number added).
  • Conversion to SciPost template, formating and sizing of figures updated.

---

## Editorial Decision

published